# A multi-pulse ultrasound technique for imaging of thick-shelled microbubbles demonstrated in vitro and in vivo

**Sigrid Berg** [1]*, **Siv Eggen**[1], **Kenneth Caidahl**[2,3], **Lars Dähne**[4], **Rune Hansen**[1]

**1** SINTEF Digital, Department of Health Research, Trondheim, Norway, **2** Department of Clinical Physiology, Karolinska University Hospital, Stockholm, Sweden, **3** Karolinska Institutet, Stockholm, Sweden, **4** Surflay Nanotec GmbH, Berlin, Germany

* sigrid.berg@sintef.no

**Data Availability Statement:** Underlying data is fully available at the following repository: https://doi.org/10.11582/2021.00072.

## Abstract

Contrast enhanced ultrasound is a powerful diagnostic tool and ultrasound contrast media are based on microbubbles (MBs). The use of MBs in drug delivery applications and molecular imaging is a relatively new field of research which has gained significant interest during the last decade. MBs available for clinical use are fragile with short circulation half-lives due to the use of a thin encapsulating shell for stabilization of the gas core. Thick-shelled MBs can have improved circulation half-lives, incorporate larger amounts of drugs for enhanced drug delivery or facilitate targeting for use in molecular ultrasound imaging. However, methods for robust imaging of thick-shelled MBs are currently not available. We propose a simple multi-pulse imaging technique which is able to visualize thick-shelled polymeric MBs with a superior contrast-to-tissue ratio (CTR) compared to commercially available harmonic techniques. The method is implemented on a high-end ultrasound scanner and in-vitro imaging in a tissue mimicking flow phantom results in a CTR of up to 23 dB. A proof-of-concept study of molecular ultrasound imaging in a soft tissue inflammation model in rabbit is then presented where the new imaging technique showed an enhanced accumulation of targeted MBs in the inflamed tissue region compared to non-targeted MBs and a mean CTR of 13.3 dB for stationary MBs. The presence of fluorescently labelled MBs was verified by confocal microscopy imaging of tissue sections post-mortem.

## Introduction

Contrast enhanced ultrasound imaging is an important diagnostic tool in clinical practice, where microbubbles (MBs) are injected intravenously to enhance the signal from the blood pool. The MBs in clinical use consist of thin lipid or protein shells encapsulating the gas core, and applications for imaging such MBs are based on harmonic methods, taking advantage of the highly flexible shell which facilitates resonant and nonlinear backscattering at low incident mechanical indices [1–5]. Pulse inversion (PI), amplitude modulation (AM) or combinations of the two are most common harmonic methods on clinical ultrasound scanners. In the later

**Funding:** SB: Liaison Committee between the Central Norway Regional Health Authority (RHA) and the Norwegian University of Science and Technology (NTNU) SB and RH: Research Council of Norway (240410/F20) SB, SE, LD, KC and RH: European commission (7th framework program, 3MiCRON (245572) project).

**Competing interests:** SB and RH are co-inventors of a patent describing the ultrasound contrast imaging method (EP3125770). LD is the founder of Surflay Nanotec GmbH, the company that produced the microbubbles used in this work.

years substantial research effort has been focused on functionalizing MBs, either by incorporating drugs for enhanced drug delivery [6–8] or attaching targeting ligands for active molecular targeting towards disease-specific biomarkers [9–11]. The shell properties of functionalized MBs may become very different from the thin-shelled MBs, and hence new imaging methods optimized for thick-shelled MBs may be required. A thick shell (>100nm) will in general have higher viscosity and therefore introduce higher damping of the bubble oscillation compared to a thin shell. Compared to ultrasound imaging of thin-shelled MBs, higher mechanical indices are typically required to drive thick-shelled MBs into nonlinear oscillations [12]. Contrast harmonic imaging methods typically rely on transmit pulses with very low mechanical indices, sufficient to invoke nonlinear scattering from thin-shelled MBs while suppressing harmonic components, as well as the strong fundamental component, from soft tissue. Tissue harmonic imaging methods, with higher mechanical indices, have been in clinical use for more than two decades [13] and the contrast-to-tissue-ratio (CTR) will typically be destroyed when increasing the mechanical index for imaging of MBs. We propose a novel multi-pulse imaging method where imaging pulses are combined with intermediate manipulation pulses at high mechanical index. Radiation force both from the imaging and manipulation pulses, in addition to possible nonlinear effects from changes in the shell result in contrast enhancement of thick-shelled MBs with adequate tissue suppression. Contrast enhancement methods involving the use of radiation force have been proposed previously, especially focusing on pushing targeted MBs towards the vessel wall to enhance the targeting effect [14–16]. The radiation force imposed by the proposed method might also enhance the targeting, but it is also the primary source of the MB detection signal.

A thicker and more stable shell can increase the in vivo stability of the MBs and increase the circulation half-life [17, 18] which currently is limited to 1–3 minutes with thin-shelled MBs [19, 20]. An increased MB circulation time is especially important for drug delivery and molecular imaging applications. For drug delivery, it will contribute to enhanced accumulation of drug in target regions, when the MBs themselves are loaded with drugs, and to prolonged duration of MB oscillations within a target region with co-injection of drugs and MBs. For molecular imaging, it will contribute to enhanced exposure of target tissues to the targeted MBs.

Molecular imaging utilizes the principle of labelling an image sensitive vector with a ligand that can bind specifically to receptors expressed by cells in the body, and the purpose is to use a non-invasive method to discover alterations in the physiology at a molecular level. Ultrasound molecular imaging can primarily be performed using targeting towards biomarkers expressed by structures within the lumen of blood vessels, such as the luminal surface of the endothelial cells, and several research groups have successfully demonstrated an increased accumulation of targeted MBs in preclinical experiments in tumor tissue [21–23], inflammation [24] and in atherosclerosis [25, 26]. BR55 (Bracco Research inc.) with specificity towards VEGFR2 is the first targeted ultrasound contrast agent that have reached clinical trials. Breast, ovarian and prostate cancer were the inclusion criteria in the first trials [27, 28], and there is ongoing recruitment on pancreatic tumors (NCT03486327 at www.clinicaltrials.gov).

As a proof-of-concept study of the novel MB imaging technique developed for imaging thick-shelled MBs, and to show their suitability for ultrasound molecular targeting, a simple and fast inflammation model suited for soft tissue is proposed. Inflammation is a basic, yet complex dynamic response of the vascularized body systems to any harmful stimuli. The classical macroscopic signs of inflammation are pain, heat, redness, swelling and loss of function. However, when studying inflammation on the microscopic level, several cascades of cellular and microvascular reactions are present, which can be exploited in targeted drug delivery and molecular imaging [29]. Induction of a sterile inflammation can be achieved by injecting

zymosan in the organ of interest. Zymosan is a substance derived from the cell wall of the yeast *Saccharomyces cerevisiae*, and has previously been used to study peritonitis [30, 31], arthritis [32], multiple organ dysfunction syndrome (MODS) [33], lung disease [34] and the effect of anti-inflammatory drugs [35]. The induction of a local inflammation by zymosan injection in muscle tissue has previously been used to investigate gamma camera imaging of inflammation [36], and a similar approach for induction of inflammation in the hind leg using lipopolysaccharide (LPS) has been reported [24].

In this paper, we present a proof-of-concept study where a novel ultrasound contrast imaging technique, which is suitable for imaging thick-shelled MBs is demonstrated. The new method is tested in vitro and in vivo, and we provide imaging results from a zymosan-induced inflammation model in muscle tissue in rabbit and compare the amount of targeted and non-targeted MBs in inflamed and healthy tissue.

## Materials and methods

### Microbubble preparation

The synthesis of the thick-shelled polymeric MBs has been described by Cavalieri et al [37, 38]. The MBs were manufactured at Surflay Nanotec GmbH, Berlin, Germany, and they have a shell made of polyvinyl alcohol (PVA) [39, 40]. These MBs are obtained by foaming a solution of PVA previously oxidized with sodium metaperiodate. The PVA chains are cross-linked during reaction occurring at the water/air interface. Resulting MBs have an average diameter of 3 μm with a shell thickness of about 200 nm and an approximate concentration of $10^9$ MB/ml. Concentration is measured by counting MBs in a Neubauer Cell under a confocal laser scanning microscope. A microscopy image of the MBs and their size distribution is shown in Fig 1.

In the molecular ultrasound imaging experiments MBs were functionalized by letting remaining aldehyde groups in the PVA shell react with Aminoguanidine hydrochloride yielding a positive charge. This enables a further layer-by-layer-coating [41, 42] by (Poly(styrenesulfonate/Poly(ethyleneimine)₂/Polystyrenesulfonate. The second Polyethylenimine layer was labeled with the fluorescent dye Tetramethylrhodamine-isothiocyanate (TRITC). These MBs with only TRITC in the shell are called *tritc-MBs*. For the second type of the MBs a further layer of biotin labelled Poly(allylamine) was coated to the surface. On the biotinylated surface streptavidin was bound, enabling attachment of biotinylated polyclonal rabbit anti-ICAM-1 (CD54) (bs-4615R-Biotin), purchased from Bioss Inc., Woburn, MA, USA. Here 60 μg of biotinylated antibody was added to 1.5 ml of MBs at a concentration of $10^9$ MBs/ml. Before

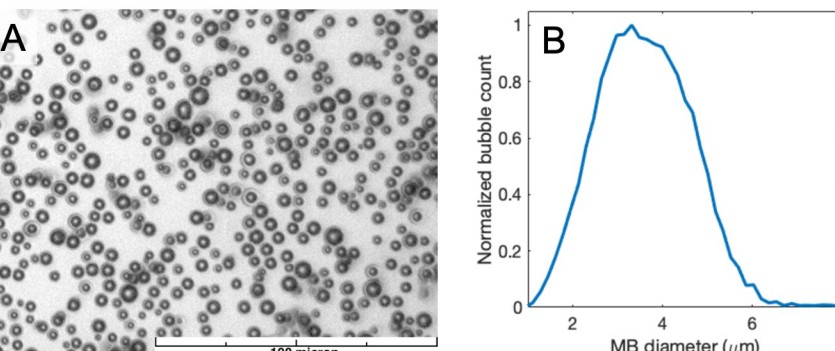

**Fig 1. Microscopy image of a typical sample of the polymeric MBs (A), and size distribution calculated based on automatic counting of optical microscopy images (B).**

injection into animals the MB solution was diluted in saline to $2 \times 10^8$ MB/ml. The MBs with both TRITC and streptavidin are called *strep-tritc-MBs*.

**In-vitro ultrasound imaging.** Due to their thick shell, the PVA MBs used in the current experiments were not well imaged by conventional harmonic imaging schemes based on PI or AM. An experimental multi-pulse technique (as described below) was therefore implemented, and imaging results from PI and AM were compared with the new technique. The novel multi-pulse technique was implemented on a GE Vivid E9 scanner (GE Vingmed, Horten, Norway) modified for research, and the 11L linear transducer was used. Comparison to clinically available techniques was also performed, and a GE Vivid E95 in clinical mode with the 9L transducer was used for PI and AM imaging.

In the multi-pulse scheme, several pulses were transmitted in each beam direction. The first and the last pulse were identical, and were used for imaging, whereas one or several intermediate pulses could be included for additional manipulation of the MBs, as illustrated in Fig 2. The intermediate pulses could have a different center frequency and number of oscillations, however, in the presented experiments the transmitted intermediate pulses were equal to the imaging pulses due to limited access to the transmit setup files of the scanner. Contrast enhanced images were formed by subtracting the echoes resulting from the imaging pulses (first and last) transmitted along each image line.

Transmit frequencies at 8 and 9.5 MHz were tested, and acquisition schemes with MI of 0.7 and 1.0, and pulse lengths of 3 and 5 half periods were compared. Images were recorded and stored in the scanner archive, and analysis on recorded images were done in the GE software EchoPAC (GE Vingmed, Horten, Norway). In addition, RF-data were recorded and processed in an in-house Matlab-environment.

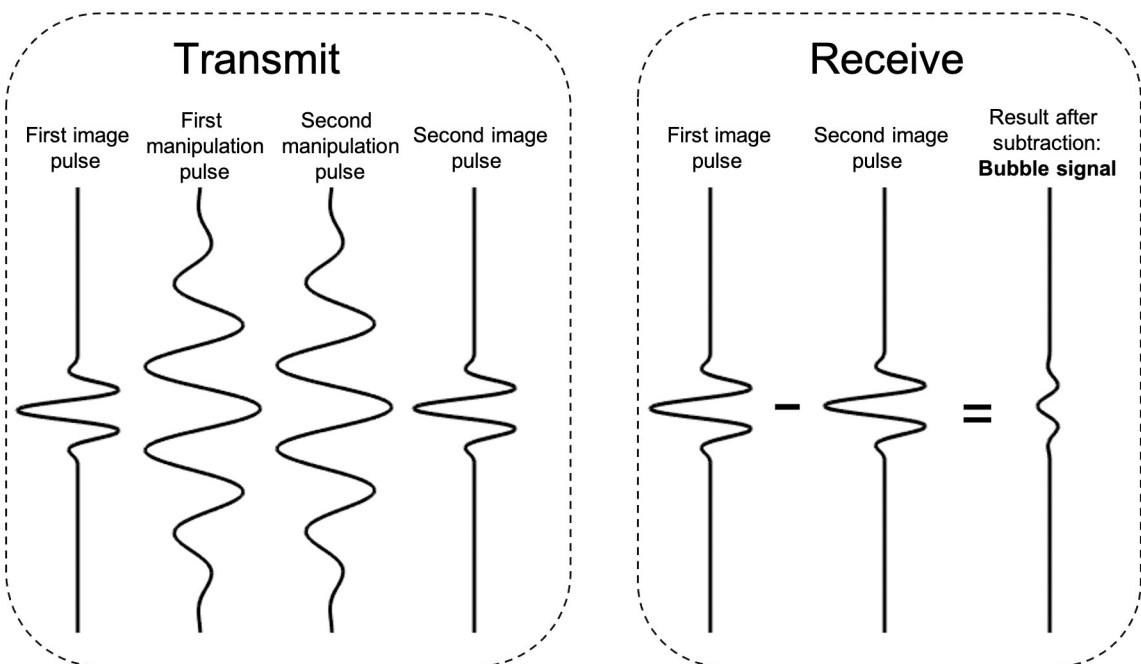

**Fig 2. Illustration of the proposed multi-pulse imaging technique.** For each scanline, two imaging pulses and several manipulation pulses were transmitted. Upon reception the last pulse was subtracted from the first, resulting in a contrast enhanced signal for that particular scanline. The pulse length, frequency and amplitude of the intermediate manipulation pulses could be different from the imaging pulses, but in this work, all transmitted pulses along one scanline were equal due to practical reasons.

Ultrasound imaging tests, method development and optimization with the thick-shelled polymeric MBs were performed with an in-vitro setup consisting of a tissue mimicking flow phantom from ATS, model 524 (ATS Laboratories, Norfolk, VA, USA) and a peristaltic pump. The ultrasound transducer was fixed with a clamp and MBs were imaged both during low-speed flow and when the pump had stopped and the MBs were stationary. All analysis of CTR was done on images recorded after the pump was stopped.

### In-vivo experiments

**Chemicals.** Zymosan A (Sigma Aldrich Co., St. Louis, MO, product no: Z4250) was suspended at 1% in 0.15 M sodium chloride and placed in a boiling water bath for one hour, followed by centrifugation for 30 minutes at 4000 rpm. The supernatant was discarded, and the residue suspended evenly in 0.9% NaCl to a concentration of 35 mg/ml.

**Animal preparation and experimental procedures.** Experimental procedures with female New Zealand White rabbits were conducted in compliance with protocols approved by the Norwegian National Animal Research Authorities (Protocol number: FOTS-5360) and all animals were acclimatized for at least one week before the experiments started. Illustrations of the timelines and experimental procedures are shown in Fig 3.

Rabbits (Hsdlf:NZW) were purchased at 15 weeks of age from Harlan Laboratories (The Netherlands). During acclimatization, the rabbits were housed as group of three in a rabbit cage system supplied by Scanbur, type EC3, and during the experimental period they were housed individually. Commercial diet and free access to hay and water were provided. They were purchased with status as specific pathogen free (SPF), but were housed in a non-SPF section since the Comparative Medicine Core Facility at NTNU does not offer SPF conditions for rabbits. The rabbits were sedated with a subcutaneous injection with a solution of fentanyl 0.2 mg/ml and fluanisone 10 mg/ml (Hypnorm®, VetaPharma Ltd, Leeds, UK) at a dosage of 0.3 ml/kg and were given an intravenous injection of midazolam (5 mg/ml (Midazolam B. Braun, Melsungen, Germany) at a dosage of 2 mg/kg) through a venflon in the lateral ear vein to

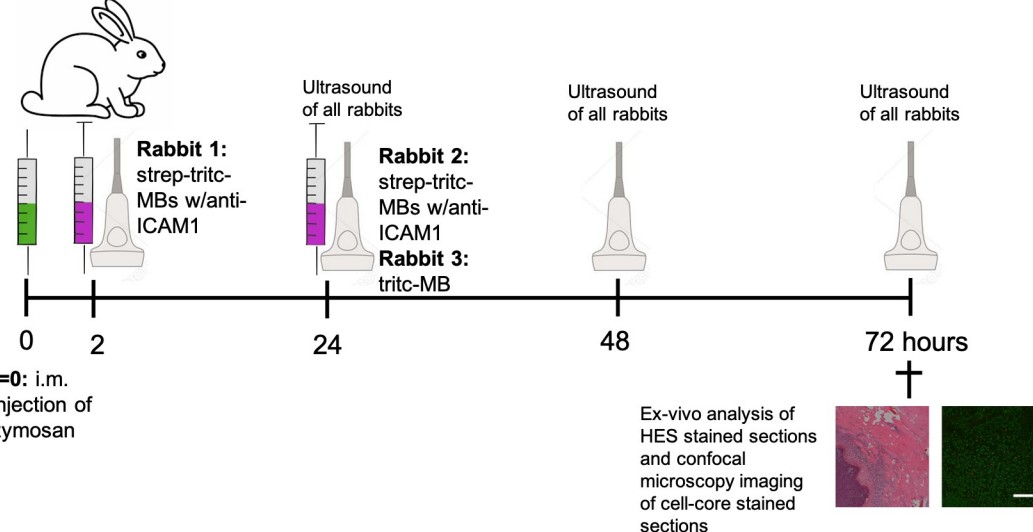

**Fig 3. Illustration of timeline and experimental procedures in the in-vivo experiments.** An intramuscular injection of zymosan was given at t = 0 (indicated by a green syringe), and MBs were administered i.v. at t = 2 hours to the first rabbit and at t = 24 hours for the second and third rabbit (indicated by a pick syringe). Ultrasound imaging was performed on all three days and all rabbits were euthanized 72 hours after the zymosan injection.

obtain full anesthesia. The anesthesia was maintained during the experiment with inhalation of 2% isoflurane gas in oxygen to keep the anesthesia at a surgical level. The venflon was kept in place for the administration of contrast agents. On the last day of experiments, the rabbits were euthanized by an intravenous administration of 5 ml of Pentobarbital 100 mg/ml.

Three rabbits were included in the experiment, and sedation and anesthesia were repeated on each day for four days. The fur on both hind legs was removed using an electrical clipper. On the first day, a sterile inflammation was induced by an ultrasound-guided intramuscular injection of 1 ml zymosan, at a concentration of 35 mg/ml in the area of the biceps femoris muscle in the left hind leg. The right hind leg was the negative control. Two rabbits were given intravenous injections of approximately $5 \times 10^8$ *strep-tritc-MBs* with anti-ICAM1 following the zymosan injection. The first rabbit was given the MBs 2h after zymosan injection, whereas the second rabbit was given the MBs 24 h after zymosan injection. The third rabbit was given an injection of approximately $5 \times 10^8$ *tritc-MBs* 24 h after the zymosan injection. The development of the inflammation and the MBs were imaged by ultrasound at 24h, 48h and 72h. All rabbits were sacrificed at 72h. Post-mortem, tissue biopsies were excised from both the left and right hind leg of all three rabbits. The samples were embedded in O.C.T. (Tissue-tek, Sakura Fine-tek, The Netherlands) snap frozen and stored at -80 degrees until sectioning.

A pain scoring sheet was logged daily for all the animals to assess their general condition and clinical symptoms. To ensure minimal pain the rabbits were given daily subcutaneous injections of 0.05 mg/kg and 0.03 mg/kg of buprenorphine (Temgesic®; Reckitt Benckiser, UK) respectively.

**Molecular ultrasound imaging.** B-mode and contrast-enhanced ultrasound images of the hind legs of the rabbits were acquired using a GE Vivid E9 scanner modified for research and an 11L linear transducer. The simple multi-pulse imaging technique was used to enhance the MBs. Images were acquired during MB injection and at 24h, 48h and 72h after the inflammation was induced. A frequency of 9.5 MHz and MI of 1 was used in the contrast images, and 12 MHz and MI of 1 was used to acquire the B-mode images. Preliminary imaging showed that the inclusion of intermediate pulses caused increased tissue signal in form of flashing. In order to minimize the flashing from tissue and increase the framerate, the in vivo recordings were done without any intermediate pulses, hence only the two imaging pulses were transmitted along each beam line. The CTR was calculated based on traces exported from the Q-analysis tool in the EchoPAC software. In each recording which was analyzed, 8 circular regions of interest (ROIs) with diameter of 0.5 mm were drawn in positions where MBs were present in some of the recorded frames, and the CTR was calculated as the difference between the peaks in signal level found when an MB was present, and the background level in the same position, when no MB was seen. All datapoints below -50 dB were considered to come from tissue and all peaks above -43 dB were considered to be caused by MBs. Multiple MBs were typically detected in each of the ROIs. Ultrasound video files used for CTR analysis and signal level traces from all ROIs are provided in the data repository.

**Histology and microscopy imaging.** Sections with a thickness of 4 μm and 25 μm were made from the rabbit muscle tissue blocks. The 4 μm sections were HES-stained and the 25 μm sections were mounted with Vectashield mounting medium with DAPI (Vector Laboratories Inc., Burlingame, CA, USA). A selection of the tissue sections was examined by microscopy techniques.

HES stained tissue sections were examined with a Nikon white light microscope with objectives from 4x magnification. Tissue sections of 25 μm with DAPI stained cell cores were examined with a confocal laser scanning microscope (CLSM) (Zeiss LSM510, Germany), where a Helium-Neon (HeNe) laser line at 543 nm was used to detect the fluorescent dye TRITC in the MBs, with emission in the range 560 nm to 615 nm, and a multi-photon Titanium-Sapphire

(TiSp) laser line at 750 nm was used to detect the DAPI bound to DNA in the cell nuclei, with emission in the range 420 nm to 470 nm. A 20x /0.5 air objective was used in combination with the tile scan function to obtain images of the whole tissue sample. Each single image had a resolution of 512x512 pixels and covered 450x450 μm. A total of 11 section from inflamed tissue and 7 sections from healthy tissue from the three rabbits were examined. MBs were automatically counted in ROIs of 1 x 1 mm$^2$ with a custom-made Matlab script (R2019a, The MathWorks Inc., Natick, MA, USA), and the number of MBs in the ROI with the maximum amount of MBs in each section were registered and stored.

## Results

### In vitro ultrasound imaging

Ultrasound image optimization performed using a tissue mimicking flow phantom showed that with the traditional PI and AM techniques, which are implemented on most commercial scanners, it was not possible to both visualize the thick-shelled MBs properly and suppress the signal from the tissue. This is shown in the panel A and B of Fig 4. With PI it was only at an MI of 0.1 that the tissue signal was adequately low, whereas for AM the tissue suppression was satisfactory also at MI of 0.2. However, the MBs were not possible to detect at either of these pressure levels. An increase of the MI to 0.6 was needed to achieve adequate MB detection, but the tissue signal at such high MI also became high. AM had better tissue suppression in shallow parts of the image, but in the deeper parts, the signal intensity from the tissue mimicking material was in the same range as the MB signal. Hence, neither of the methods were suitable for imaging the thick-shelled MBs in small blood vessels and capillaries. The dynamic range was the same for all PI images, but the gain was adjusted for each MI setting to get a similar level of electronic noise within the flow tube at all three pressure levels.

Images acquired with the multi-pulse detection method proposed in this work are shown in the two lower panels of Fig 4. In a stationary phantom, the subtraction of the second imaging pulse from the first, resulted in images where the tissue signal was completely suppressed. When only the two imaging pulses were used, the signal from the MBs was clearly seen inside the flow channel, but including one or two intermediate manipulation pulses between the imaging pulses, enhanced the MB signal even more. In the images which included manipulation pulses the improved detection can be seen by a somewhat higher intensity and an increased size of the MBs. The tissue signal is not affected by the manipulation pulses in this setup.

When recording RF data from the scanner, an analysis of the received pulses along each scanline is possible. An example of RF data analysis from a recording at 9.5 MHz with an MI of 1.0 and pulse lengths of 3 half periods is shown in Fig 5. The backscattered signal from many of the MBs is much lower than the strongest tissue signal, as can be seen in the B-mode image in Fig 5A, which is based on the first imaging pulse only. Subtracting the last pulse from the first results in a contrast enhanced image which is shown in Fig 5B. When comparing the amount of MBs in the image in Fig 5A and 5B, it is apparent that not all MBs become visible with this method. However, the MBs that are detected have a high CTR. In Fig 5C–5F, the MB marked with a white arrow in Fig 5B is further analyzed. The received imaging pulse along the scanline through the MB is displayed in Fig 5C, and a zoom-in on the backscattered signal from the MB is shown in Fig 5D. Subtracting the last imaging pulse from the first, results in a subtraction signal where the stationary areas of the scanline are close to zero and the signal from the MB is enhanced due to a small translation as shown in Fig 5E (and zoomed in in Fig 5F). Analysis of the signal from this particular MB gives a detected delay between first and last pulse of 5 ns, corresponding to a translation of 3.9 μm and a resulting CTR of 22.6 dB.

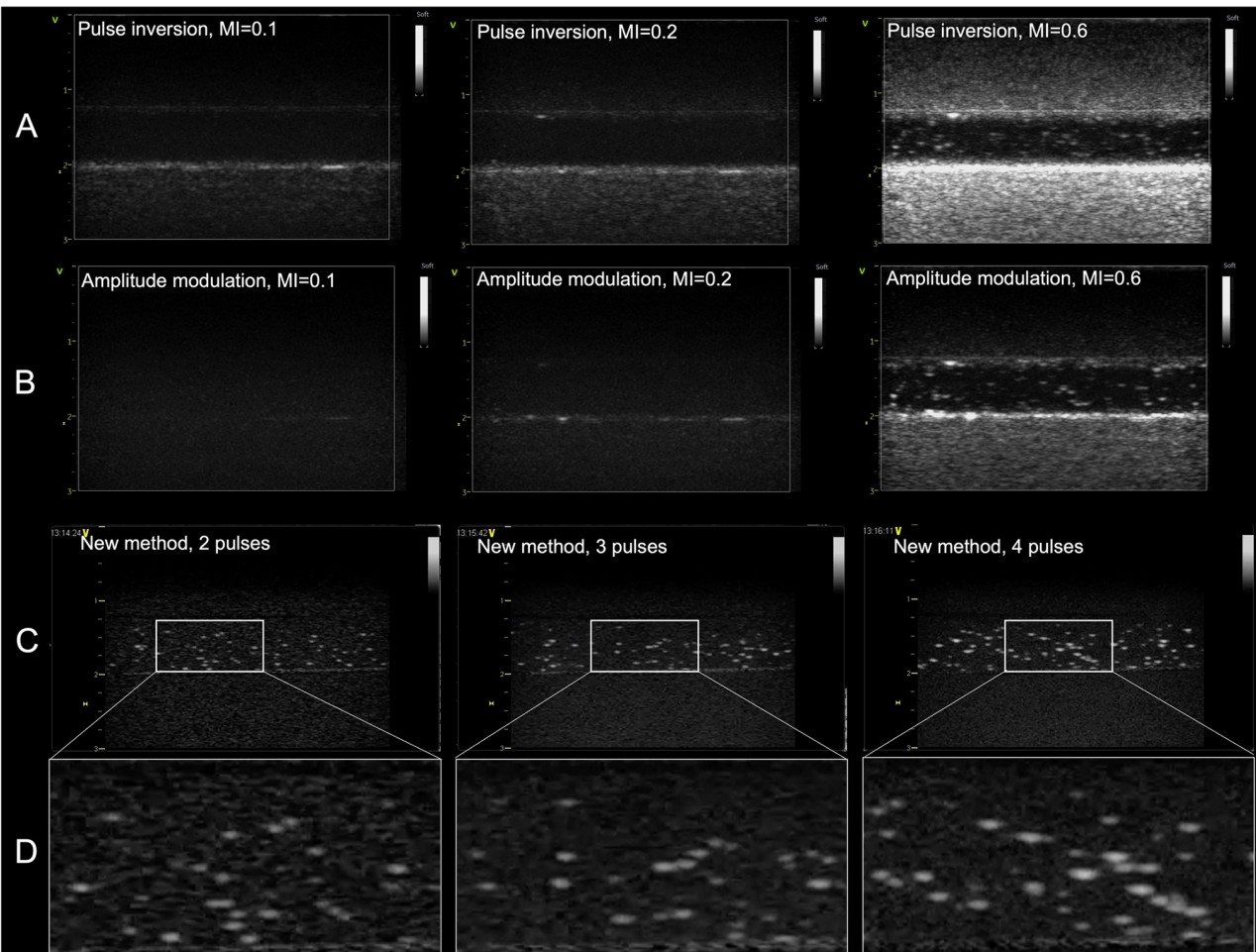

**Fig 4. Ultrasound images of MBs in a tissue mimicking flow phantom.** Panel A shows PI images with transmit at 3.2 MHz and receive at 6.4 MHz and panel B shows AM images recorded at 7 MHz, both are at MIs of 0.1, 0.2 and 0.6. Images were recorded with the GE Vivid E95 system and the 9L linear transducer and with a dynamic range of 54 dB. Panel C shows examples of the new method with zero, one and two intermediate pulses respectively, recorded with an 11L linear transducer and the GE Vivid E9 system. The MI is 1.0 for all examples and a frequency of 12 MHz and dynamic range of 30 dB was used. In panel D a region of the flow channel is enlarged to show how detection improves when one or two intermediate manipulation pulses are included in the image acquisition.

A systematic collection of RF data from images recorded with two intermediate pulses at 8 and 9.5 MHz, with MI of 0.7 and 1 and pulse lengths of 3 and 5 half periods was analyzed. When subtracting the second imaging pulse from the first, the delay caused by a translation of the MBs of 2–5 μm resulted in a CTR of up to 23.1 dB. The backscattered signals from the 6 to 10 brightest MBs in each image were analyzed, and the detected mean delay between the first and last imaging pulse along the scanline through the MB and the mean CTR are presented in Table 1. In addition to the results in Table 1, RF data from images recorded at 8 MHz without manipulation pulses was also recorded. At an MI of 1 the mean delay when using 3 hp and 5 hp imaging pulses was 2.2 ± 0.7 ns and 2.3 ± 0.8 ns respectively. The corresponding mean CTR was 13.0 ± 1.9 dB and 21.7 ± 2.1 dB for 3 and 5 hp. At MI = 0.7 there were hardly any visible MBs in the contrast image.

The high MI setting (MI = 1.0) was the most important contributor to an increased CTR, with a 7–10 dB increase compared to recordings at MI of 0.7 and otherwise equal settings.

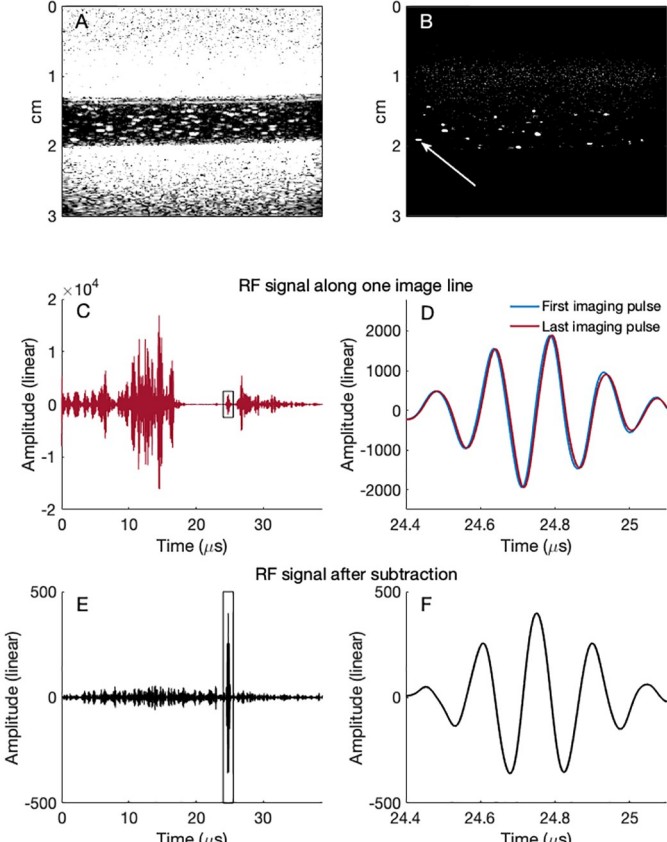

**Fig 5. A representative example of the signal from a MB in a tube in a tissue mimicking phantom.** The analysis is based on RF data from in-vitro imaging with a GE Vivid E9 scanner. A: B-mode image of stationary MBs in a flow channel through a tissue mimicking phantom. The image is based on only the first imaging pulse. B: Contrast enhanced image resulting from subtracting the last imaging pulse from the first. The image line which goes through the center of the MB marked by the white arrow is analyzed further. C: Received signal from imaging pulses along the indicated image line. D: Received signal from the MB marked by the white arrow (zoomed in on the signal marked by a black box in panel C). The second pulse is delayed by approximately 5 ns compared to the first pulse. E: The resulting signal when the second imaging pulse is subtracted from the first. F: Resulting MB signal after subtraction (zoomed in on the data marked by a black box in panel E).

Pulse length and frequency are of less importance. In order to get the best image resolution, the highest frequency and shorter pulse length were chosen as initial settings for the in vivo experiments, i.e. 9.5 MHz and 3 half periods. However, due to motion, and flashing artefacts when using the intermediate manipulation pulses in vivo, the in vivo results were recorded without manipulation pulses at MI = 1.

**Table 1. Mean delay and CTR ratio in images recorded at 8 and 9.5 MHz.** Two intermediate manipulation pulses between the imaging pulses were used in the experiments.

| Frequency | 8MHz | | | | 9.5MHz | | | |
|---|---|---|---|---|---|---|---|---|
| Pulse length | 3 hp | 5 hp | 3 hp | 5 hp | 3 hp | 5 hp | 3 hp | 5 hp |
| MI | 1.0 | | 0.7 | | 1.0 | | 0.7 | |
| Mean delay ± std [ns] | 5.0 ± 0.9 | 5.3 ± 0.6 | 2.6 ± 0.4 | 2.4 ± 0.5 | 4.4 ± 0.7 | 4.2 ± 0.5 | 2.2 ± 0.4 | 2.4 ± 0.3 |
| Mean CTR ± std [dB] | 23.1 ± 2.0 | 20.4 ± 3.2 | 16.0 ±3.2 | 13.9 ± 3.3 | 22.4 ± 2.5 | 19.2 ± 1.7 | 12.9 ± 1.9 | 11.7 ± 1.6 |

Abbreviations: contrast-to-tissue ratio, CTR; half period, hp

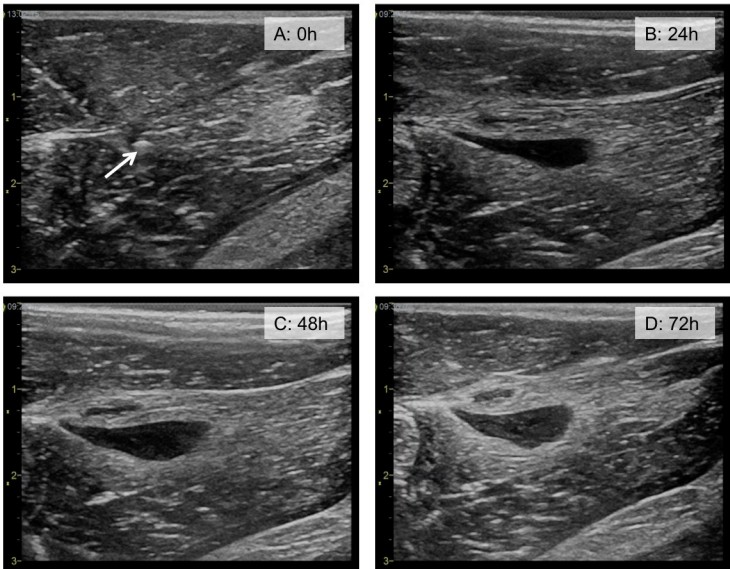

**Fig 6. Ultrasound B-mode images from Rabbit 1.** Image a) shows healthy muscle tissue, and the white arrow points to the tip of the needle injecting the zymosan. Images b), c) and d) shows the hypoechoic region, which developed over the three days following the zymosan injection. The imaging depth is 3 cm and dynamic range is 69 dB.

## In-vivo ultrasound imaging

B-mode ultrasound images of healthy thigh muscles generally showed a quite uniform signal intensity from the muscle tissue. In the thigh muscle with induced inflammation, a hypoechoic (dark) region was evident 24 hours post-injection of zymosan. The extent of architectural changes was different between animals, but the hypoechoic areas detected in one leg was recognizable the following days and a general observation was that these areas became more extensive with time. As shown in Fig 6 the hypoechoic (dark) area was clearly detectable after 24 hours and increased in size after 48 hours and 72 hours. Upon injection of MBs, the circulating MBs were detected by the novel multi-pulse contrast enhanced ultrasound imaging method in both small and large vessels within the thigh muscle for several minutes. CTR analysis of the recording from each of the rabbits just after the MB injections resulted in a tissue signal level of -53.1 ± 2.0 dB and contrast signal of -36.3 ± 4.6 dB, giving a CTR of 16.8 dB for flowing MBs. Ultrasound videos of inflow of MBs are available in the data repository. When imaged 24h, 48h and 72h after the injection, no circulating MBs were seen; however, a large number of *strep-tritc-MBs* with anti-ICAM-1 were detected along the perimeter of the inflamed area (Fig 7D). By contrast, the *tritc-MBs* without active targeting were not found close to the inflammation, but some *tritc-MBs* were detected within the areas of healthy muscle tissue (Fig 7F). No apparent MB destruction was observed by ultrasound imaging, even though the MI during imaging was 1.0. The signal from the non-circulating MBs was stable over the entire imaging period, and CTR analysis of one recording at each timepoint for each of the rabbits resulted in the same signal level from tissue as before and an average contrast signal of -39.8 ± 2.5 dB, giving a CTR of 13.3 dB for static MBs. A total of 172 local signal peaks representing MBs were found in the ROIs from images recorded just after MB injection and 205 local peaks were found in the ROIs in images 24h, 48h, and 72h after MB injection. Figures showing examples of the traces representing signal within selected ROIs and the distribution of CTR levels from all the detected MBs from the various recordings can be found as S1 File.

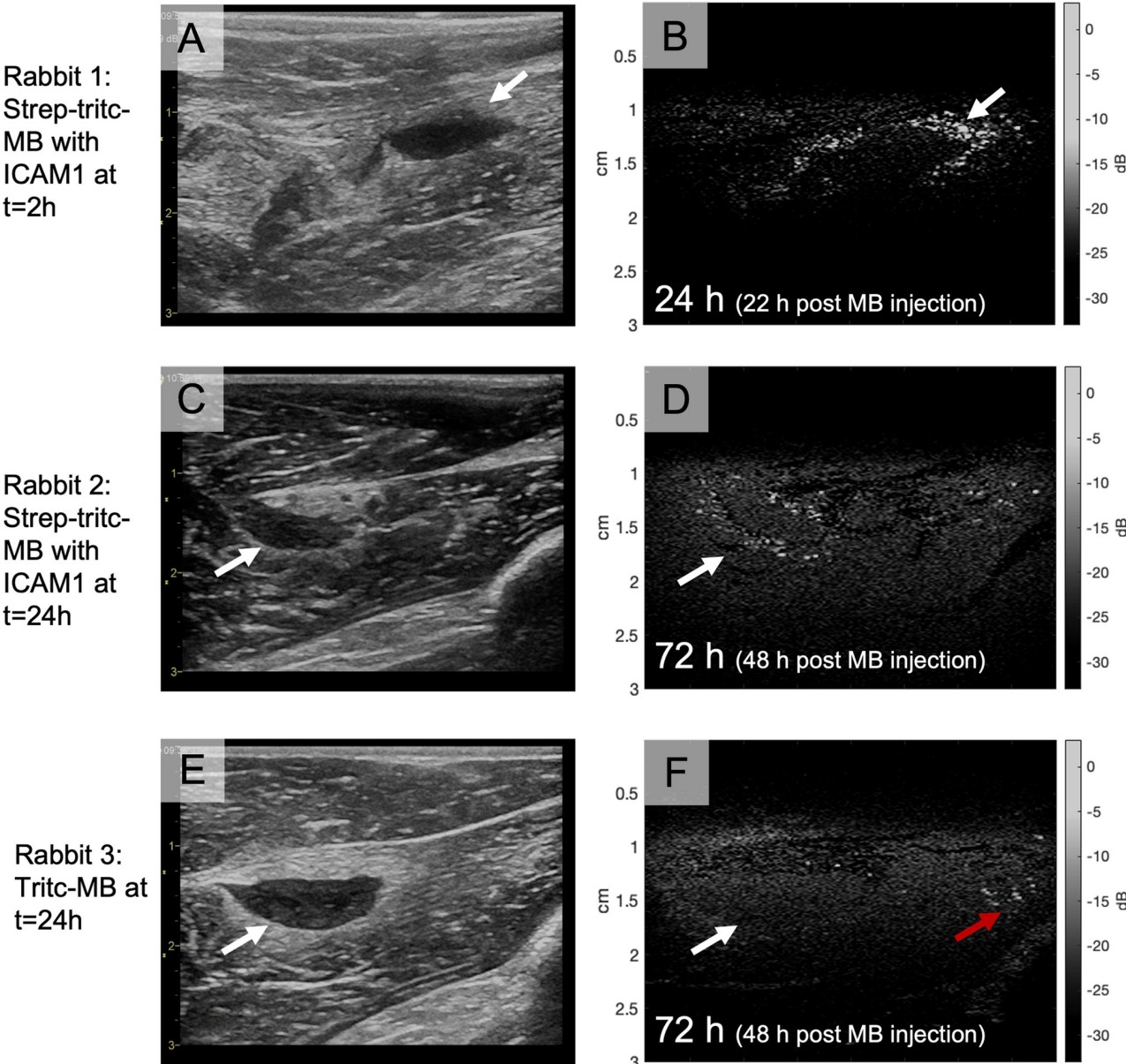

**Fig 7. Examples of B-mode (A, C and E) and contrast enhanced (B, D and F) images from the three rabbits included in the study.** Images A and B are from Rabbit 1, which was injected with strep-tritc-MBs with anti-ICAM1 2 hours after the onset of the inflammation. The images shown are recorded 22 hours after MB injection, and the area of inflammation is marked with white arrows. Images B and C are from Rabbit 2, which was injected with strep-tritc-MB with anti-ICAM1 24 hours after the inflammation was induced, and the images are recorded 48 hours after MB injection. The inflammation area and accumulation of MBs at its perimeter are marked by white arrows in the images. Some MBs are also detected further from the inflammation (right side of (D). In images E and F examples from Rabbit 3 are shown. The animal was given tritc-MB (MBs without streptavidin and antibodies) 24 hours after the inflammation was induced, and the images are recorded 48 hours after MB injection. The contrast enhanced image shows some MBs in the muscle tissue at the far-right end (red arrow), but no accumulation of MBs around the inflammation area (white arrow). B-mode images are recorded at 12 MHz and shown with a dynamic range of 69 dB and contrast enhanced images are recorded with imaging pulses of 9.5 MHz and a dynamic range of 36 dB.

**Muscle tissue necropsy.** Upon post-mortem dissection, an accumulation of a pale, yellow viscous exudate was found in the inflammatory region. The tissue in the parts of the muscle surrounding the inflammation was darker in color, hence indicating that the blood flow was enhanced as a sign of an inflammatory reaction.

**Ex vivo analysis.** In HES-stained tissue sections, evident changes could be seen in the muscle tissue from the hind leg where the inflammation had been induced, in all animals. The accumulation of inflammatory cells was extensive and infiltrated areas between the surrounding muscle fibers, as seen in Fig 8A. The exudate found in the tissue with inflammation consists of a protein-rich fluid, dead leukocytes and cellular debris. The inflammatory cells consisted of mainly granulocytes, but also mononuclear cells were present. In areas of the tissue sections located 3–5 mm away from the accumulation of inflammatory cells, the muscle tissue morphology appeared to be normal.

The confocal microscopy examination of tissue sections from the inflamed and healthy muscle tissue from rabbits that were given *strep-tritc-MBs* with anti-ICAM-1 and *tritc-MB* without targeting confirmed the findings from the contrast enhanced ultrasound images. A large number of MBs were found in the areas close to the pus-filled regions compared to the areas further away. In the inflamed muscle tissue of the rabbit that was given *tritc-MBs* without any active targeting and in the healthy muscle tissue, hardly any MBs were detected by confocal microscopy imaging. An example image from the rabbit that was given targeted *strep-tritc-MBs* 2h after induction of inflammation is shown in Fig 8. Enhancement of an area within the active inflammation (Fig 8C) and an area 1 mm away from the inflammation (Fig 8D) show that MBs are mainly found in close vicinity to the pus-filled region. The MBs can be seen as red dots in Fig 8C.

MBs were counted in a total of 11 tissue sections from inflamed tissue and 7 sections from healthy tissue, and the results are shown in Fig 9. The highest number of MBs was found in the inflamed tissue from the rabbit which was given the strep-tritc-MBs with anti-ICAM1 only two hours after the inflammation was induced. In the healthy tissue the mean number of MBs/ mm$^2$ varied from 0 to 1.3, and no apparent difference was found between targeted and non-targeted MBs.

**Animal health.** The animals did not display any visible signs of pain or distress, and the body weight did not change significantly during the experimental period. The inflammation was not detectable by visual inspection only, but through palpation, areas that were firmer than a healthy thigh muscle were found in the areas where the zymosan had been injected.

## Discussion

In this study, we have demonstrated a simple multi-pulse technique for imaging thick-shelled MBs both in vitro and in vivo in an inflammation model for soft tissue, where its suitability for use in molecular ultrasound imaging also is shown.

### Ultrasound imaging technique

When comparing the proposed multi-pulse technique with the commercially available PI and AM techniques, we showed that the multi-pulse technique was superior in visualizing thick-shelled MBs. PI utilizes the fact that scattering from commercially available thin-shelled MBs, like SonoVue, will be nonlinear even at very low mechanical indices (below 0.1) when the driving frequency is below or in the vicinity of the MBs resonance frequency. For MBs with a thick shell or a less flexible shell, and when the driving frequency is above the resonance frequency, higher mechanical indices are required to obtain adequate nonlinear scattering from the MBs. Higher mechanical indices will result in nonlinear wave propagation through the tissue causing harmonic components to be backscattered from tissue. With harmonic detection techniques, the differentiation between MBs and tissue will then be compromised, as was shown in the upper panels of Fig 4. AM could be expected to perform better than PI when imaging thick-shelled MBs, since it relies on the difference in MB response between two pulses with

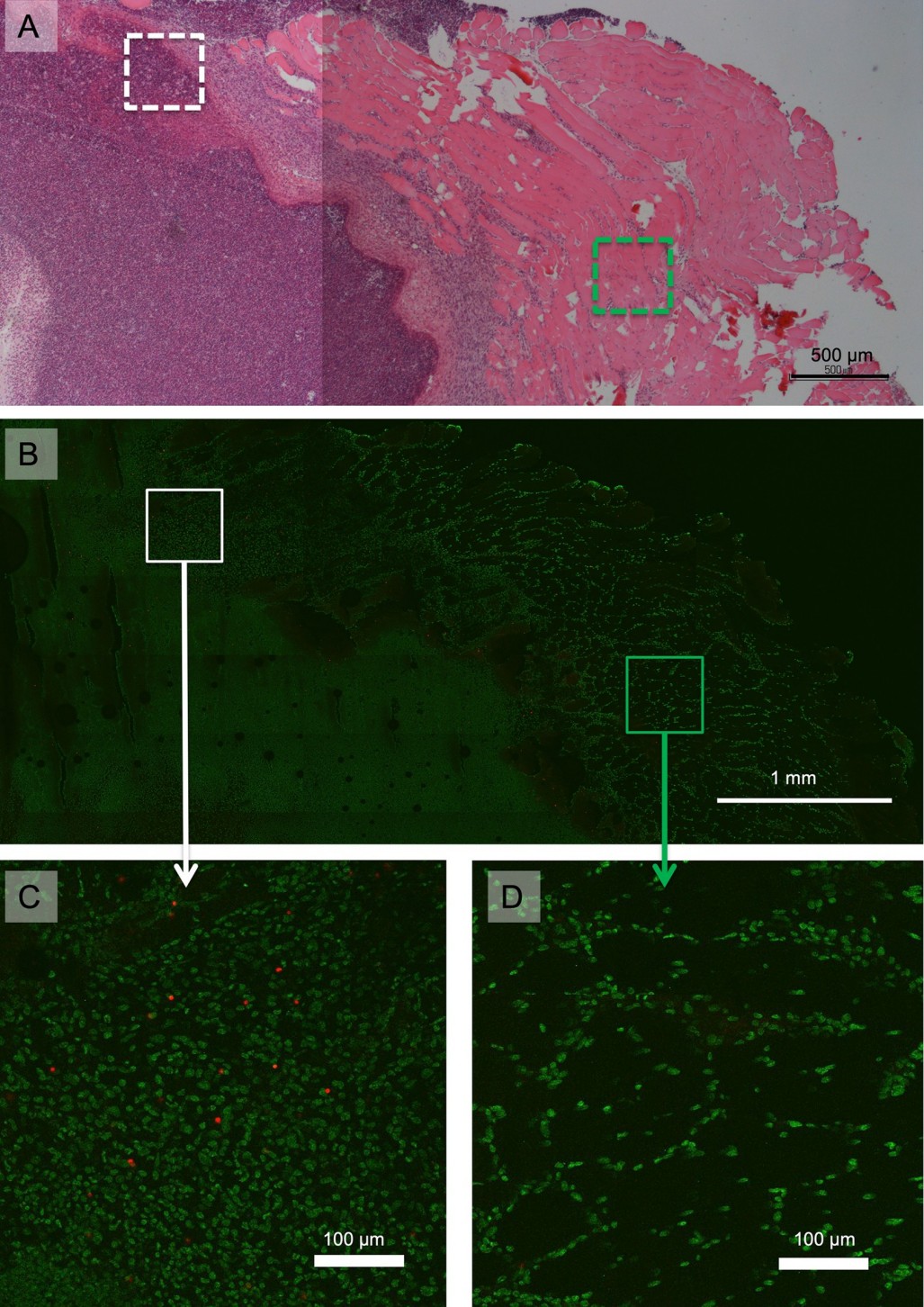

**Fig 8. Example of HES (A) and corresponding DAPI mounted (B-D) tissue sections from the inflamed tissue from the rabbit that was injected with strep-tritc-MBs with anti-ICAM-1 2h after inflammation was induced and sacrificed 70h later.** Confocal microscopy images (B, C and D) are from a 25 μm thick tissue section. The cell nuclei are stained with DAPI and shown in green and the MBs containing TRITC are shown in red. Two selected regions (0.5x0.5mm) from the confocal image (B) are displayed in detail in C and D. The white boxes represent areas of pus-filled region displayed in C, and the green boxes mark an area approximately one millimeter away from the pus-filled region and contains muscle tissue with only slight infiltration of inflammatory cells. Boxes with full lines represent the area in the confocal image and corresponding positions in the HES-stained section are marked with dashed boxes.

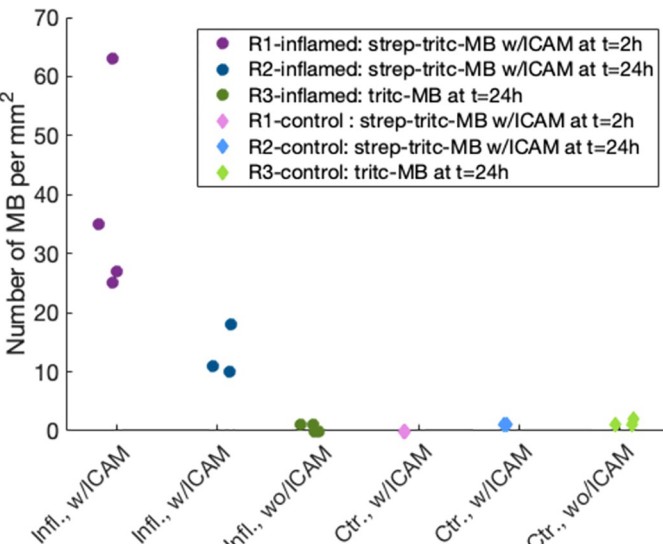

**Fig 9. Number of MBs found in a 1mm x 1 mm region of interest (ROI) in 25 μm thick tissue sections from inflamed (dark colors) and healthy (light colors) muscle tissue from rabbits.** The circular markers are from sections of inflamed tissue, and diamond markers are from sections from the contralateral healthy side. Pink and blue markers represent injections of strep-tritc-MBs with anti-ICAM, and green represent tritc-MBs without any targeting. Each tissue section that was examined is represented by one data point, a total of 11 sections from inflamed tissue and 7 sections from healthy tissue were examined.

different amplitude [43]. When imaging thin shelled MBs this difference is caused by MB oscillation. In a situation with thick-shelled MBs one could envision that a high amplitude pulse could cause a backscatter signal, e.g. due to shell buckling [44], whereas the low amplitude pulse did not, and by that detecting the presence of MBs. However, the in vitro imaging showed that both PI and AM only could detect MBs at MI levels which were too high for the tissue suppression to work properly, whereas there was no detectable MB signal at low MIs.

When using the multi-pulse technique, the MB-signals arise due to a difference in the received signals between the two imaging pulses transmitted along each scan line. The most important contribution to the detection signal appears to be a delay between the two resulting backscattered signals due to a small physical translation of the MBs caused by the radiation force from the incoming pulses. In the example shown, the detected delay was in the range 2–5 ns, which corresponds to a translation of a few micrometers. In the in vitro setup, the tissue mimicking material is stationary and the MBs are in a tube with a diameter much larger than the MB diameter. In this idealized situation a CTR of up to 23 dB was achieved. In an in vivo situation, the tissue is not expected to be stationary, and the MBs in small vessels or capillaries will have limited ability to undergo a translational motion. Hence the CTR was expected to be lower. In the example of MBs imaged in the vicinity of the inflammation area a mean CTR of 13.3 dB was obtained.

Including several manipulation pulses or a manipulation pulse of longer duration between the imaging pulses may contribute to detection of a larger fraction of MBs and stronger signal from the MBs as long as the MBs are able to physically translate, but prolonged time between the two imaging pulses will also result in increased tissue signal unless the tissue is stationary. Tissue motion may especially be a problem in the abdominal and thoracic areas, where breathing and the beating heart may cause strong artefacts if no motion compensation is included in the implementation. In the current work, a relatively stationary organ was imaged (the thigh).

Artefacts due to motion were therefore minimized and the use of motion compensating algorithms was not required, however, there is still some flashing effects which may be caused by probe translation and can be seen in the videos in the data repository.

Another potential contribution to the detection signal is due to a change in amplitude or pulse form between the two resulting signals scattered from the MBs. Changes in amplitude or pulse form may be caused by a partial rupture, shell perturbation or deflation of the MB due to the oscillations caused by the incident pressure pulses. We have not been able to verify the presence of such effects in the current work, but the same kind of MBs have been studied with high speed optical imaging during ultrasound exposure, and it was found that at low frequencies (2–4 MHz) the shell buckled and that the gas was pumped out of the MBs when bursts of 10 cycles were applied [45]. However, several consecutive pulses at high MI were needed to effectively rupture the MBs, and the pulses used in the presented experiments are only 3–5 half cycles, and at a much higher frequency (9.5 MHz), so the effect on the MBs might be somewhat different than what was observed in the experiments of Kothapalli et al. A full destruction or deflation of the MB during the insonation of short imaging pulses is unlikely, and also backed by the observations during the imaging.

The multi-pulse technique proposed in this study is implemented with a transmit frequency at 8 and 9.5 MHz but can in principle be used at any frequency range with suitable transducers and ultrasound scanners. A modified detection sequence based on power Doppler using short (e.g., 1-cycle) pulses of suitable ensemble length, for optimal spatial resolution and clutter/tissue filtering, with intermediate manipulations pulses of suitable frequency and duration, for inducing bubble decorrelation, is probably the optimal implementation of this method. This will be pursued in future projects.

**The inflammatory reaction.**   The accumulation of pus in the region where the zymosan was injected is a sign that leukocytes have been recruited from the circulation and transmigrated into the tissue where the initial damage occurred. This recruitment is a result of production of cytokines and subsequent inflammatory specific changes in the endothelium that allows for circulating immune cells to enter the injured area and, if possible, eliminate the cause and repair the damage. The ultrasound images show that the onset of the inflammatory reaction was rapid, and that the amount of pus and extent of infiltration of inflammatory cells in the muscle tissue increased with time. It is a great advantage, both for animal health and cost of experiments, that the time from induction of disease to the point where the molecular imaging can be performed is short.

**Molecular ultrasound imaging.**   The polymer MBs used in these experiments have longer blood circulation time compared to commercially available lipid MBs [46], and the ultrasound images of MBs in the inflamed area show that single MBs can be detected even days after injection. Active targeting and hence attachment of MBs to endothelial cells might hinder or diminish the translation effect due to radiation force and it will probably also reduce the fraction of MBs being detected. However, in most situation, a component of the radiation force will act in the flow direction of the capillary, which could cause the MBs to detach from the endothelial cells. During imaging we observed that the detected MBs could disappear after being imaged for some time (typically ranging from about 5 seconds to several minutes) potentially indicating some sort of rolling and eventual detachment.

Studies of similar PVA MBs containing superparamagnetic iron oxide (SPIONs) in the shell as MR imaging contrast agent, showed that the MBs were phagocytized by macrophages, mainly in the liver and spleen, and that the process took weeks [17, 18]. This is also supported by the work of Härmark et al [46], where PVA MBs were found only inside vasculature, but often taken up by or in the vicinity of macrophages. Successful targeting of ICAM-1, VCAM-

1, and e-selectin using layer-by-layer MBs has been shown in human and murine endothelial cells and in a zymosan induced peritonitis model [42].

Results from the histological examination show that the rabbit given *strep-tritc-MBs* with anti-ICAM-1 only 2 hours after induction of the inflammation had a higher concentration of MBs in the vicinity of the pus-filled regions compared to the rabbit given the injection 24 hours after inflammation induction. This indicates that the total circulation time (70 hours vs 48 hours) resulted in a higher number of MBs attaching to biomarkers. Another possible reason is that more targeted biomarkers were available 2 hours post induction than at 24 hours after the inflammation was induced. However, with the limited number of animals included in the proof-of-concept study it is not possible to draw a conclusion in this matter. The *tritc-MBs* without anti-ICAM-1 were not found in the vicinity of the inflamed region, indicating that the active targeting with anti-ICAM-1 was the main cause for the high number of MBs close to the inflammation. However, since the *tritc-MBs* did not contain streptavidin in the shell, the possibility of an unspecific binding of the streptavidin in the *strep-tritc-MBs* to the inflamed regions cannot be excluded. The *strep-tritc-MBs* detected by ultrasound in healthy tissue 24h and 48h after injection may also indicate a level of a general unspecific binding due to the streptavidin on the MB shell. Some *tritc-MBs* were also detected within healthy tissue in the contrast enhanced ultrasound images. However, based on confocal microscopy evaluation of tissue section the number of MBs within normal tissue was very low for all three rabbits. The increased blood flow in the inflamed area may also contribute to the high number of MBs found, compared to normal muscle tissue.

## Conclusion

A simple multi-pulse imaging technique for contrast enhancement of thick-shelled MBs has been described and implemented on a high-end ultrasound scanner. Comparison to traditional PI and AM techniques showed superior in vitro detection quality. In vivo, the technique was demonstrated in molecular ultrasound imaging of a soft tissue inflammation model in rabbit. MBs targeted towards the inflammatory marker ICAM-1 were injected and visualized in the muscle tissue upon injection and up to 70 hours after injection. An increased number of MBs in the perimeter of the inflamed area was demonstrated both on contrast enhanced ultrasound images and on histological examinations of excised tissue.

The presented soft tissue inflammation model enables molecular imaging of a relevant inflammatory biomarker within a day after the induction of inflammation and does not lead to apparent pain or discomfort for the animals. It is a flexible model with respect to both anatomy and species, which makes it highly relevant for testing and optimization of new contrast agents and contrast agent imaging techniques.

## Supporting information

**S1 File.**
(DOCX)

## Acknowledgments

The procedures involving laboratory animals were performed at the Comparative Medicine Core Facility (CoMed), Norwegian University of Science and Technology (NTNU), Trondheim, Norway. The embedding of tissue samples, preparation of tissue sections, HES staining was performed at the Cellular & Molecular Imaging Core Facility (CMIC), NTNU. Thanks to dr. Per Stenstad at SINTEF Industry, Trondheim, Norway for attaching antibodies to the MBs,

and to GE Healthcare for allowing us to implement an experimental contrast imaging scheme on the GE Vivid E9 system.

## Author Contributions

**Conceptualization:** Sigrid Berg, Siv Eggen, Kenneth Caidahl, Lars Dähne, Rune Hansen.

**Formal analysis:** Sigrid Berg.

**Funding acquisition:** Kenneth Caidahl, Rune Hansen.

**Investigation:** Sigrid Berg, Rune Hansen.

**Methodology:** Sigrid Berg, Siv Eggen, Kenneth Caidahl, Rune Hansen.

**Project administration:** Rune Hansen.

**Resources:** Siv Eggen, Lars Dähne.

**Supervision:** Rune Hansen.

**Visualization:** Sigrid Berg.

**Writing – original draft:** Sigrid Berg, Siv Eggen.

**Writing – review & editing:** Siv Eggen, Kenneth Caidahl, Lars Dähne, Rune Hansen.

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
