## [Decision Letter · Decision Letter 0]

2 Feb 2022

PONE-D-21-28949A multi-pulse ultrasound technique for imaging of thick-shelled microbubbles demonstrated in vitro and in vivoPLOS ONE

Dear Dr. Berg,

Thank you for submitting your manuscript to PLOS ONE. After careful consideration, we feel that it has merit but does not fully meet PLOS ONE’s publication criteria as it currently stands. Therefore, we invite you to submit a revised version of the manuscript that addresses the points raised during the review process.

Three expert reviewers provided detailed and relevant comments. Among many other issues, some "solvable" flaws were identified. It is clear that the framework of the comparative evaluation with different sequences is seriously questioned since the acoustic pressure used and frequencies are different. Additional experiments would be required to better support your statements and quantitative analysis would also be required in vivo. Dr Cloutier

We look forward to receiving your revised manuscript.

Kind regards,

Guy Cloutier, Ph.D.

Academic Editor

PLOS ONE

Journal Requirements:

SB: Liaison Committee between the Central Norway Regional Health Authority (RHA) and the Norwegian University of Science and Technology (NTNU) 

SB and RH: Research Council of Norway (240410/F20) 

SB, SE, LD, KC and RH: European commission (7th framework program, 3MiCRON (245572) project)

 SB and RH are co-inventors of a patent describing the ultrasound contrast imaging method (EP3125770). LD is the founder of Surflay Nanotec GmbH, the company that produced the microbubbles used in this work.

Additional Editor Comments:

Three expert reviewers provided detailed and relevant comments. Among many other issues, some "solvable" flaws were identified. It is clear that the framework of the comparative evaluation with different sequences is seriously questioned since the acoustic pressure used and frequencies are different. Additional experiments would be required to better support your statements and quantitative analysis would also be required in vivo.

Reviewers' comments:

Reviewer's Responses to Questions

**Comments to the Author**

1. Is the manuscript technically sound, and do the data support the conclusions?

Reviewer #1: Yes

Reviewer #2: Partly

Reviewer #3: Partly

2. Has the statistical analysis been performed appropriately and rigorously? 

Reviewer #1: N/A

Reviewer #2: No

Reviewer #3: N/A

3. Have the authors made all data underlying the findings in their manuscript fully available?

Reviewer #1: Yes

Reviewer #2: Yes

Reviewer #3: Yes

4. Is the manuscript presented in an intelligible fashion and written in standard English?

Reviewer #1: Yes

Reviewer #2: Yes

Reviewer #3: Yes

5. Review Comments to the Author

Reviewer #1: The authors present a succinct piece of work using a new imaging sequence that is tailored for stiff microbubbles. The new imaging sequence is very similar to amplitude modulation, in that it is measuring the difference between the response of a bubble population over time, which can be due to translation, bubble rupture, or other effects. I would have liked to see a direct comparison between the new sequence and AM, not PI as the authors did. The comparison between PI, indeed, was taken at entirely different frequencies and pressure amplitudes, and I’m not sure it’s a fair comparison. The work seemed to be conducted robustly, and in that regard the paper deserves eventual publication. I would perhaps suggest to the authors that, in addition to my comments below, the authors comment on the similarly between AM, PI and their new sequence since the physics isn’t different. This approach essentially is trying to get away from bubble vibration at all (since they don’t vibrate very well at low MI) and is trying to focus on signal ‘difference’ due to motion, etc. What is the expected frequency content of the resulting signal, would it work best on big or small bubbles, would it also work on lipid bubbles, etc.

Line 30: “In the later years […] I might suggest adding a reference here (like a review article or something, perhaps a Klibanov one?)

Line 33: “less resonant”. I think you’ll need to clarify this. What is a less resonant bubble? You mean it’s resonant oscillation magnitude is smaller than that of a resonant lipid-coated bubble of the same size and shell properties? I’m not sure that’s such an easy comparison but I do see your point. Please clarify

Line 35: “Compared to ultrasound imaging of thin-shelled MBs, higher mechanical indices are typically required to drive thick-shelled MBs into nonlinear oscillations.”.

• Please provide a reference

• It’s generally understood that thick-shelled bubbles need to break to elicit sufficient nonlinear oscillations (i.e. destructive nonlinear behaviour). Is this what the authors are referring to?

Line 38: “[…] while supressing harmonic components from soft tissues.” These pulses are also designed to supress fundamental signal from soft tissues, too. Please modify sentence to account for this.

Microbubble preparation: Were these measured by Coulter Counter also? Microscopy methods will not capture bubbles below 1 um (biased result). How confident are the authors in this size distribution (Fig 1)? A concentration of 1e8 is quite low, which might be a symptom of missing the smaller bubbles.

In-vitro testing: Clearly the comparison made between the novel pulse sequence and traditional pulse-inversion isn’t valid, since the MI’s are not the same and neither are the frequencies! I’m a little confused: If the authors used PI at an MI equal to that of their new sequence (MI=1), they surely would obtain usable contrast-images. Is this true? For example, the ‘new method, 2 pulses’ is actually sort of similar to AM (since there is no intervening pulses), but taken at MI=1.

In fact, this novel scheme works essentially as amplitude modulation. The idea is that the intervening pulses either move or distort the bubbles (but not tissue), and then upon subtraction of two pulses, you are left with ‘nonlinear fundamental’ signal – that is signal at the fundamental (which is linear) due to the presence or absence of a target. I’d like to see how this new sequence compares with traditional AM (which is almost the same as ‘new method, 2 pulses’).

Reviewer #2: In this manuscript entitled “A multi-pulse ultrasound technique for imaging of thick

shelled microbubbles demonstrated in vitro and in vivo”, the authors report a multi-pulse subtraction imaging technique, with interleaved manipulation pulses, to image thick shelled polymer microbubbles in vitro in a flow loop and in vivo in a rabbit model of inflammation. Their in vitro experiments are conducted using very low concentration of MB in a commercially available flow phantom. Their in vitro results indicate that their non-destructive subtraction imaging scheme, which, on responding microbubbles, can result in a CTR of 23dB, which is good. This is attributed to a translation of the MB of up ~4um, which is roughly the diameter of one MB. The manipulating pulses can theoretically be modified (pressure, length, frequency) or even completely removed, which they did for their in vivo experiments.

In vivo, they used a modified version of their imaging approach (without manipulation pulse) for the molecular imaging of ICAM1-targeted MB using and anti-ICAM1 decorated MB using biotin streptavidin conjugation, which is an adequate conjugation scheme for a proof of principle study. They demonstrate that MB can be detected at 24h when MB are injected at 2h post injury and 72h when injected at 24h post injury but not with control MB. This is supported by histological fluorescence imaging in N=1 rabbit per condition.

Overall, this reports contains some interesting data but (1) suffers from unsupported statements and inadequate experiments in vitro and (2) lacks adequate quantitative analysis in vivo (essentially N=1 and non-analysis of longitudinal data)

MAJOR FLAWS

1- For instance, since their imaging application is targeted MB, it is not acceptable to only characterize MB in flowing conditions in vitro. Especially since their approach relies on MB translation caused by radiation force. Indeed, one has to wonder if that translation will be hindered once MBs are attached. In vitro experiments with targeted MB should be performed. How many MB will respond ? Will they roll with successive pulses? Or detach ?

2- Also, in the first part of the manuscript, the authors optimize the manipulation pulses in vitro but end up using a different imaging scheme in vivo without manipulation pulses. It would be relevant to repeat the in vitro experiments with the conditions used in vivo (no manipulation) to allow comparison and understand the importance of the manipulation pulses.

3- Subtraction methods are known to be sensitive to motion. Yet their images (cf Fig 7) clearly do not reflect this reality, which is clearly visible in the in vivo supplementary videos. This limitation should be disclosed and discussed. Why is there so much movement artifact with rabbit 3 ? How does it affect the CTR ?

4- Why are only a fraction of MB responding to this imaging scheme (Figure 5)? Please quantify that. The reviewer wonders if it is fair to quantify only responding MB in table 1. Since you report results on 6-10 MB in table 1, how did you choose which MBs to analyze ? There are clearly more than 10 MB in Fig4. How did you account for the MBs that do not give signal with your method (cf Fig 5 A and B)?

5- I have never seen PI harmonic imaging with such poor tissue cancellation performance (cf Figure 4). Was this contrast package the commercial package from GE or was it home-made ? Please specify in the methods. Also, an additional in vitro image with a lipid MB, perhaps in supplementary data, would be appropriate to convince the readers of the proper coding of the PI sequence.

6- Moreover, I strongly encourage the authors to use and report the same dynamic range in all images. This is an important parameter that largely affects the visual rendering of CTR

7- Although this is a proof of principle study, reporting data with N=1 is never a good approach. Perhaps one way to bring more robustness to the study would be to analyze the images longitudinally at different time points (48h vs 72h), which you state you have performed in the methods, but are not reporting or quantifying. For Rabbit 1, you could even do 24h, 48h and 72h. L461: replace “days” by “48 hours in one animal”.

8- Is it possible that your inflammation model has some variability (hard to know with N=1) ? Could Rabbit 3 have less inflammation than rabbit 2 explaining the change in brightness and histology ? Could you analyse ICAM1 expression using histology ?

9- The selections of the analyzed sections in the histology analyses need to be shown. It is unclear how the 11 areas in the inflamed tissues and 7 areas in control muscle were selected.

10- How were in vivo CTR values computed ? How did you choose MB signal vs tissue signal ? In which frame in the video ? Please describe the method and report mean, standard deviation, #MB analyzed and how they were chosen.

11- It would be interesting to stain for endothelium to confirm the compartment of the microbubbles after 48h in vivo. Are they in the vasculature ? Uptaken by leucocytes ?

MINOR

You mention imaging during MB injection (L204) but don’t show images. Since two B-mode images are used to compute the contrast image, using one for B_mode would allow perfect coregistration and allow anatomical correspondence in Figure 7.

Please verify the excitation wavelength for your fluorophores L217-L218

Figure 7 : Perhaps using the first imaging pulse as the B-mode would help coregister B-mode and contrast images. Right now you are selecting an arbitrary frame from the cineloop than is not in the same position as the B-mode

Figure 7E : Correct label on the right side : It should be 24h correct ?

Figure 8 : Use arrows to indicate the different tissues (including the puss filled region and normal muscle tissue) and indicate features described in the text (inflammatory cells, dead leukocytes, cellular debris, granulocytes). Explain why these two sections were selected.

Reviewer #3: This manuscript describes a contrast pulse strategy to interlace 2 modifications pulses between 2 identical imaging pulses for the specific application of thick-shell targeted microbubbles. While such method can’t pick-up non-linear echoes like Pulse Inversion, it should be sensitive to agent motion and disruption. The ability to detect bounded microbubbles in-vivo 22 hours after injection is impressive. My main concern is that the authors use identical pulses for the imaging and modification components in this manuscript which is very close to a Doppler sequence (identical to a Doppler sequence if the PRF is constant). The in-vivo section uses 2 pulses which is identical to a (2-pulse) Doppler sequence. Doppler does work at detecting bubbles at high MI regimes but is not novel. While I understand the research limitation of the clinical system, my suggestion to the authors is to either design a simple single transducer in-vitro experiment demonstrating the advantage of the intermediate pulses (e.g. much longer pulse, different frequency, etc.) to distinguish the method from Doppler, or rephrase the manuscript as using Doppler for targeted agent which by itself could be novel. In the later case, measuring the Doppler spectrum of the targeted agents and comparing it to tissue would be relevant.

The introduction is also missing significant background on existing pulse sequences. Amplitude modulation in particular is relevant to this work as its application has been demonstrated to gas vesicle which closely resemble thick shell microbubbles. It is also routinely available on clinical ultrasound systems.

Comments:

(introduction) ‘Thick-shelled MBs can have improved circulation half-lives, incorporate larger amounts of drugs for enhanced drug delivery or facilitate targeting for use in molecular ultrasound imaging. However, methods for robust imaging of thick-shelled MBs are currently not available.” Comment: What does the author consider thick shell? Would Sonazoid count as a stiff shell (compared to Sonovue/Definity)?

(introduction) The introduction does not provide information about the proposed novel imaging technique. Also, the background on general pulse sequence literature is missing from the introduction.

(p.5) ‘MBs used in the current experiments were not well imaged by conventional harmonic imaging schemes based on pulse inversion’ Comment: What about amplitude modulation (AM)? In particular, one would suspect that AM would perform well for thick shell agents due to their buckling dynamics (see Marmottant model)

(p.6) The description of the pulse sequence comes really late in the manuscript and is hidden with other experimental details. I would move it closer to the introduction.

(p.6) ‘first and the last pulse were identical, and were used for imaging, whereas one or several intermediate pulses could be included for additional manipulation of the MB’. Comment: If the first and the last pulses are identical, then the pulse sequence is not extracting any nonlinear echo information but instead will pick-up either motion, bubble disruption or partial bubble disruption. Furthermore, here, both the imaging and intermediate pulses are identical which means the authors are simply using a Doppler sequence (in particular if the PRF is constant). This would need clarification in the manuscript.

(p.7 l.137) ‘and acquisition schemes with MI of 0.7 and 1.0’. Comment: While I expect hard shell bubbles to be more resilient to pressure, those are very high MI when compared to conventional contrast imaging. For traditional non-destructive imaging the norm is MI~0.06-0.1. The introduction made is sound like the contrast sequence needs to be non-destructive. This should be clarified in the manuscript.

(p.7) ‘The ultrasound transducer was fixed with a clamp and MBs were imaged both during low speed flow and when the pump had stopped and the MBs were stationary.’ Comment: Were results computed for each condition or combined? Flow introduce motion between pulses and yield a significant increase in the CTR which may shadow other effects from the intermediate pulses.

(p.10) ‘A frequency of 9.5 MHz and MI of 1 was used in the contrast images, and 12 MHz and MI of 1 was used to acquire the B-mode images. In order to minimize tissue signal and increase the framerate, the in vivo recordings were done without any intermediate pulses, hence only the two imaging pulses were transmitted along each beam line. Comment: In this case, the contrast sequence is simply a 2 pulse Doppler sequence. Doppler sequence do work at imaging bubbles, but only if they are moving are destroyed by the imaging pulse. Doppler was actually the original detection scheme for microbubbles (before PI and AM were invented), and has been documented in literature. I recommend that the authors discuss this in the introduction.

(p.12) ‘When comparing the amount of MBs in the image in Figs 5A and 5B, it is apparent that not all MBs become visible with this method’. Comment: Could this indicate that the high MI has detrimental effect on the thick shell microbubbles or that microbubble coalesce into larger bubbles that become easier to detect?

(p.13) ‘The backscattered signals from 6 to 10 MBs in each image were analyzed, and the detected mean delay between the first and last imaging pulse along the scanline through the MB and the mean CTR are presented in Table 1.’ Comment: This CTR algorithm seems biased. Bubbles are supposed to be everywhere in the flow channel; a more robust metric would be to compare larger ROI in the tube and tissue.

(p.14) ‘The CTR of circulating MBs immediate after injection was 15-20 dB.’ Comment: It would be relevant to create a new figure with the Contrast image a several time stamps after the injection in order to justify the 15-20dB claim.

(Discussion) It should be discussed why the authors didn’t consider Amplitude modulation, which is the most commonly used contrast pulse sequence on clinical systems. It is also worth noting that gas vesicles (nanobubbles with thick shells) were successfully imaged with AM due to their buckling dynamics [reference: Acoustic Behavior of Halobacterium salinarum Gas Vesicles in the High-Frequency Range: Experiments and Modeling, E. Cherin et al., UMB, Vol 43, Issue 5, pp.1016-1030] and is therefore a promising candidate for their thick shell microbubbles.

(p.19 l.428) ‘Tissue motion may especially be a problem in the abdominal and thoracic areas, where breathing and the beating heart may cause strong artefacts if no motion compensation is included in the implementation. In the current work, a relatively stationary organ was imaged (the thigh).’ Comment: This is precisely the same limitation of Doppler imaging. Doppler mitigates this issue by using a wall-filter which rejects low Doppler frequencies. I recommend to measure the Doppler spectrum from these microbubbles and compare it to the in-vivo Doppler spectrum of tissue. One would expect the microbubble spectrum bandwidth to broaden compared to tissue Doppler due to signal decorrelation.

(p.19 l.437) ‘We have not been able to verify the presence of such effects in the current work’. Comment: That being said, it is unlikely that the acoustic pulses moved the bounded bubbles in the in-vivo experiment.

Editorial

(p.5 l.87) ‘[…] has been described earlier’ Comment: Please rephrase as ‘has been described by Cavalieri et al […]’

6. PLOS authors have the option to publish the peer review history of their article (what does this mean?). If published, this will include your full peer review and any attached files.

Reviewer #1: No

Reviewer #2: No

Reviewer #3: No

---

## [Author Response · Author response to Decision Letter 0]

8 Apr 2022

We thank for the thorough review done by all three reviewers. We have addressed all the comments and answered the remarks in the attached document. Changes made in the manuscript are commented with reference to line numbers in the revised version of the manuscript.

---

## [Decision Letter · Decision Letter 1]

2 May 2022

PONE-D-21-28949R1A multi-pulse ultrasound technique for imaging of thick-shelled microbubbles demonstrated in vitro and in vivoPLOS ONE

Dear Dr. Berg,

Thank you for submitting your manuscript to PLOS ONE. After careful consideration, we feel that it has merit but does not fully meet PLOS ONE’s publication criteria as it currently stands. Therefore, we invite you to submit a revised version of the manuscript that addresses the points raised during the review process.

We look forward to receiving your revised manuscript.

Kind regards,

Guy Cloutier, Ph.D.

Academic Editor

PLOS ONE

Additional Editor Comments:

Unfortunately, one reviewer still believes that another round of revision is necessary. He is insisting on the use of amplitude modulation in new sets of experiments for comparison. I will wait for your new responses and changes before finalizing the decision.

Sincerely,

Dr Cloutier

Academic Editor

Reviewers' comments:

Reviewer's Responses to Questions

**Comments to the Author**

1. If the authors have adequately addressed your comments raised in a previous round of review and you feel that this manuscript is now acceptable for publication, you may indicate that here to bypass the “Comments to the Author” section, enter your conflict of interest statement in the “Confidential to Editor” section, and submit your "Accept" recommendation.

Reviewer #1: All comments have been addressed

Reviewer #2: All comments have been addressed

Reviewer #3: (No Response)

2. Is the manuscript technically sound, and do the data support the conclusions?

Reviewer #1: (No Response)

Reviewer #2: Yes

Reviewer #3: Partly

3. Has the statistical analysis been performed appropriately and rigorously? 

Reviewer #1: N/A

Reviewer #2: N/A

Reviewer #3: Yes

4. Have the authors made all data underlying the findings in their manuscript fully available?

Reviewer #1: (No Response)

Reviewer #2: Yes

Reviewer #3: Yes

5. Is the manuscript presented in an intelligible fashion and written in standard English?

Reviewer #1: Yes

Reviewer #2: Yes

Reviewer #3: Yes

6. Review Comments to the Author

Reviewer #1: The authors do an adequate job at replying to the reviewer's commentary. I will just clarify one thing: The authors state that AM relies on nonlinear behaviour and their proposed method relies on radiation force and is thus mechanistically different. Typically, when we refer to nonlinear bubble vibrations we are referring to their nonlinear content (e.g. subH), and I agree that in this context (thin versus thick shell), the stated difference above is reasonable since thick-shelled bubbles don't vibrate very easily. However, the power of AM comes from the 'nonlinear' signal remaining in the fundamental band due, not specifically to nonlinear vibrations, but to a non-proportional response to two input pressures (at the fundamental - i.e. linear band). This is due to the fact that a given bubble may not vibrate at all at one pressure amplitude A, and then vibrate with a very large amplitude at a transmit pressure 2A - more than just a proportional amount. This is not a nonlinear oscillation issue, it's a lack/presence of microbubble issue. In this sense, the proposed technique works very, very similarly (if not identically) to AM, as was mentioned by both reviewers.

Reviewer #2: Thank you for addressing most of my comments.

- I am not certain I understand your response to Question 1 about additional in vitro experiments:

"Unfortunately, we do not have the needed infrastructure for conducting such experiments at this point."

What infrastructure are you referring to ? Targeting MB in vitro is not that difficult. For example you can look at PMID: 19411212.

- I also think that you could do a better job at describing and quantifying your CTR in vivo results L351-356. Since you have 172 + 205 MB you have analyzed at different time points, it would be interesting to describe these results in more detail using a figure.

There are still a few typos that may require your attention

L252 : Grammar : satisfactory is an adjective. In this phrasing your should use an adverb such as "adequately" ? or perhaps write : "was low, as expected"

L313: Grammar : ... in Table 1. "Additionally to the results in Table 1," RF data from images...

L319 : In table1 : I believe that a period is missing for MI=0.7 3HP standard deviation

L343: Please add : 16.8dB "for flowing MB"

L353 : Please add : 13.3 dB "for static MB"

L378 : you say 30 dB but the color bar is suggesting 35 dB. Please verify.

L383 : Is is common to display DAPI in blue. I understand this is pseudo colouring but it is confusing when looking at the image. If possible please use blue for DAPI.

L516 : Grammar : please add a period and cut in two sentences : ... [14,15]. This is also supported by...

L518 : replace "towards" with "of"

Reviewer #3: I would like to thank the author for revising the manuscript. However, I still believe it would be worthwhile to properly investigate the advantage of the manipulation pulse both in-vitro and in-vivo. While the modulation has theoretical advantages over power Doppler, this is not quite demonstrated in the manuscript. I also think it would be worthwhile to compare the proposed sequence to Amplitude modulation in a flow phantom setup (as done for PI).

(l. 326) “However, due to motion, and flashing artefacts when using the intermediate manipulation pulses in vivo, the in vivo results were recorded without manipulation pulses at MI=1.”

Comment: The addition of the manipulation pulse will decrease the pulse repetition frequency between the imaging pulses, but should also increase the bubble displacement and improve detection from background tissue. Similar to Power Doppler, the detectability of the microbubbles will be tied to the amount of displacement detected relative to surrounding tissue displacement.

(From reviewer response) “Gas vesicles typically have very thin protein shells (approx. 2nm or less) and the vesicle itself has a cylindrical dimension with diam of 50-100 nm and length of 100-1000 nm. The PVA bubble has a thick polymer shell of approx. 200 nm. Our GE Vivid scanner did not have an implementation of AM and there is no reason to believe that AM will work adequately for thick-shelled PVA bubbles. The buckling-effect and compression-only behavior for example seen with SonoVue (having a lipid shell of approx. 2 nm) is unlikely with a thick polymer shell of approx. 200 nm.”

Comment: While it is true that gas vesicle differs from thick shell microbubbles, there is experimental and theoretical evidence in literature that thick shell microbubbles will buckle through reversible collapse. In particular see [P. Marmottant et al. Buckling resistance of solid shell bubbles under ultrasound, JASA, 129 (3), 2011]. As the CTR achieved by AM is strongly affected by the buckling dynamics of bubbles, there is reason to believe that AM will perform much better than PI [ref: C. Tremblay-Darveau, IEEE Trans UFFC, 65 (8), 2018]. It seem highly relevant, and the authors should ideally perform an in-vitro validation of their technique to AM (which should be readily available on most clinical systems) even if done on a different clinical system.

(From reviewer response) “ One might argue that the technique has similarities with conventional Doppler but Doppler techniques are made for imaging an existing flow, not to induce a translation due to ultrasound radiation force as with our method. That is why we want to use imaging pulses tailored to the organ size and depth of interest and intermediate manipulation pulses that are tailored to maximize the radiation force on the PVA bubbles. Also, Doppler makes use of transmit pulses consisting of several pulse oscillations (resulting in reduced range resolution) and several pulse packets needs to be transmitted in each beam direction for spectral analysis (to construct an adequate clutter filter for suppressing strong, slow moving scatterers and to estimate blood velocities).”

Comment: I agree that pulses used for CW and PW are typically quite long, but Colour or Power Doppler pulses are typically of length comparable to that of Contrast pulses. If the manipulation pulse is taken out (as done in the in-vivo section), then the pulse sequence is equivalent to a N=2 ensemble power Doppler sequence (and is not novel). It would be worthwhile to try the power Doppler mode on the GE machine (using the lowest PRF) and see how it compares to the proposed pulse sequence. It is also not clear to me why the manipulation pulse is taken out for the in-vivo imaging. The authors claim that using the manipulation pulse causes too much tissue motion clutter (l.326), but clutter is a fundamental limitation of any Doppler system. Does this mean that the implementation of the manipulation pulse is not practical in-vivo? I understand the theoretical advantages of using a long burst for the manipulation pulse, but this is not demonstrated in-vivo in the manuscript.

(From reviewer response) “The aim of the current paper was not to describe various contrast pulse sequences and reference 3 provides a good overview of such sequences”

Comment: The reader should be able to infer from the introduction where the authors work sits within the existing literature. For instance, is there any literature work describing how PI/AM/AMPI do not work well with hard shell microbubble? Any theorical work on solid shell microbubble that could be use to justify the pulse sequence (Marmottant’s paper on stiff shell buckling dynamic seems relevant)? If the goal of the modulation pulse is to displace microbubble, some citation of papers describing acoustic radiation forces on bubbles is also relevant.

(From reviewer response) “ (p.19 l.437) ‘We have not been able to verify the presence of such effects in the current work’. -> That being said, it is unlikely that the acoustic pulses moved the bounded bubbles in the in-vivo experiment. -> We agree. Knowing exactly what the cause of the detection signal from each MB in an in-vivo situation is impossible (except for maybe in intra-vital microscopy at very high magnification and frame rate). In vitro, it is possible to study single bubble behavior, as has been done in Kohtapalli at al (2015).”

Comment: This limitation should be described in the Discussion section, including possible hypothesis from existing literature.

7. PLOS authors have the option to publish the peer review history of their article (what does this mean?). If published, this will include your full peer review and any attached files.

Reviewer #1: No

Reviewer #2: No

Reviewer #3: No

---

## [Author Response · Author response to Decision Letter 1]

29 Jun 2022

We thank for the second round of thorough review done by all three reviewers. We have addressed the comments and answered the remarks in the document included in the submission. Changes made in the manuscript are commented with reference to line numbers in the revised version of the manuscript. Images showing a comparison of the novel method to both Amplitude Modulation and Pulse Inversion is now included in the manuscript, as requested.

---

## [Decision Letter · Decision Letter 2]

4 Aug 2022

PONE-D-21-28949R2A multi-pulse ultrasound technique for imaging of thick-shelled microbubbles demonstrated in vitro and in vivoPLOS ONE

Dear Dr. Berg,

Thank you for submitting your manuscript to PLOS ONE. After careful consideration, we feel that it has merit but does not fully meet PLOS ONE’s publication criteria as it currently stands. Therefore, we invite you to submit a revised version of the manuscript that addresses the points raised during the review process.

Your manuscript has been assessed again by the three reviewers from the previous rounds, who are satisfied that their concerns have been addressed.  However, when carrying out our editorial assessment of this manuscript, we noted that your data availability statement states that some underlying data cannot be shared because of existing non-disclosure agreements.  PLOS journals require authors to make all data necessary to replicate their study’s findings publicly available without restriction at the time of publication - restrictions on data availability because of commercial reasons such as non-disclosure agreements do not comply with this policy (see full details of the PLOS data policy at https://journals.plos.org/plosone/s/data-availability). To comply with the PLOS data policy, please ensure data underlying all of your findings are made available, outlining in your data availability statement how the data can be accessed. Alternatively, if it is not possible to make all data available, please remove the parts of the manuscript for which the raw supporting data cannot be provided and include a detailed justification in your response-to-reviewers that outlines which data were removed and if/how your conclusions are affected. We will not be able to proceed further with your manuscript until these concerns are addressed.

We look forward to receiving your revised manuscript.

Kind regards,

Dr Joseph Donlan

Senior Editor

PLOS ONE

Journal Requirements:

Reviewers' comments:

Reviewer's Responses to Questions

**Comments to the Author**

1. If the authors have adequately addressed your comments raised in a previous round of review and you feel that this manuscript is now acceptable for publication, you may indicate that here to bypass the “Comments to the Author” section, enter your conflict of interest statement in the “Confidential to Editor” section, and submit your "Accept" recommendation.

Reviewer #1: All comments have been addressed

Reviewer #2: All comments have been addressed

Reviewer #3: All comments have been addressed

2. Is the manuscript technically sound, and do the data support the conclusions?

Reviewer #1: (No Response)

Reviewer #2: Yes

Reviewer #3: Yes

3. Has the statistical analysis been performed appropriately and rigorously? 

Reviewer #1: (No Response)

Reviewer #2: Yes

Reviewer #3: Yes

4. Have the authors made all data underlying the findings in their manuscript fully available?

Reviewer #1: (No Response)

Reviewer #2: Yes

Reviewer #3: Yes

5. Is the manuscript presented in an intelligible fashion and written in standard English?

Reviewer #1: (No Response)

Reviewer #2: Yes

Reviewer #3: Yes

6. Review Comments to the Author

Reviewer #1: (No Response)

Reviewer #2: Thank you for addressing my concerns.

In the end, the quality of your in vivo results convinced me that this should be published.

Reviewer #3: Thank you for including the new AM data to the manuscript and for the detailed responses. I have no additional comment.

7. PLOS authors have the option to publish the peer review history of their article (what does this mean?). If published, this will include your full peer review and any attached files.

Reviewer #1: No

Reviewer #2: No

Reviewer #3: No

---

## [Author Response · Author response to Decision Letter 2]

7 Sep 2022

After communication with representatives from GE Healthcare, we were allowed to also share the IQ-data which in the first versions were not included in the original submission. This is the data that forms the basis of Table 1 and Figure 5 in the manuscript. The IQ data is included in the data repository as .mat-files.

The manuscript is re-submitted unchanged.

---

## [Editor Report · Decision Letter 3]

5 Oct 2022

A multi-pulse ultrasound technique for imaging of thick-shelled microbubbles demonstrated in vitro and in vivo

PONE-D-21-28949R3

Dear Dr. Berg,

We’re pleased to inform you that your manuscript has been judged scientifically suitable for publication and will be formally accepted for publication once it meets all outstanding technical requirements.

Kind regards,

Dr Joseph Donlan

Senior Editor

PLOS ONE
---

## [Editor Report · Acceptance letter]

12 Oct 2022

PONE-D-21-28949R3 

A multi-pulse ultrasound technique for imaging of thick-shelled microbubbles demonstrated in vitro and in vivo 

Dear Dr. Berg:

I'm pleased to inform you that your manuscript has been deemed suitable for publication in PLOS ONE. Congratulations! Your manuscript is now with our production department. 

Kind regards, 

on behalf of

Dr Joseph Donlan 

Staff Editor

PLOS ONE